# Effect of COVID-19-Induced Stress among Males on the Partner Relationship and Sexual Activity during COVID-19 Pandemic: A Cross-Sectional Study

**DOI:** 10.3390/healthcare10091663

**Published:** 2022-08-31

**Authors:** Meshari A. Alzahrani, Mohammad Alkhamees, Sulaiman Almutairi, Saad M. Abumelha, Muhammad Anwar Khan, Zainab Y. AL-Jaziri, Fay A. Althunayyan, Basel O. Hakami, Mohammad Shakil Ahmad

**Affiliations:** 1Department of Urology, College of Medicine, Majmaah University, Al Majmaah 11952, Saudi Arabia; 2Division of Urology, Department of Surgery, Ministry of the National Guard-Health Affairs, Riyadh 11481, Saudi Arabia; 3King Abdullah International Medical Research Center (KAIMRC), Riyadh 11426, Saudi Arabia; 4College of Medicine, King Saud Bin Abdulaziz University for Health Sciences (KSAU-HS), Riyadh 11426, Saudi Arabia; 5College of Medicine, Saud bin Abdulaziz University for Health Sciences (KSAU-HS), Jeddah 22384, Saudi Arabia; 6King Abdullah International Medical Research Centre (KAIMRC), Jeddah 22384, Saudi Arabia; 7College of Medicine, King Faisal University, Hofuf 31982, Saudi Arabia; 8Department of Urology, King Faisal Medical City for Southern Region (KFMC), Abha 62527, Saudi Arabia; 9Department of Family & Community Medicine, College of Medicine, Al Majmaah 11952, Saudi Arabia

**Keywords:** COVID-19, stress, sexual relationship, partner relationship, psychosexual health

## Abstract

Introduction: Since the onset of the coronavirus disease 2019 (COVID-19) pandemic, there have been some reports regarding the impact of COVID-19 on male psychosexual health. Aims and Objectives: To assess the severity of stress during COVID-19 and to determine the association of stress levels with partner relationships and sexual activity. Methodology: A cross-sectional study was conducted in Saudi Arabia through social media platforms via an online questionnaire between 1 December 2020 and 31 January 2021 among 871 participants after a pilot study among 20 participants, of which 497 were included in the study. Stress levels were assessed using the Arabic version of the Depression, Anxiety, and Stress Scale (DASS-21). Statistical analysis was conducted using SPSS version 20.0. Responses were presented as frequencies and percentages, and associations were studied using the Chi-squared test/Fisher’s exact test. A value of *p* ≤ 0.05 was considered significant. Results: A total of 497 participants who had been infected with COVID-19 completed the survey. In total, it was found that 203 (40.8%) had severe stress scores (severe and extremely severe scores merged), while 131 (26.4%) had moderate stress scores. About 84 (16.9%) participants agreed that their sexual desire decreased, 91 (18.1%) confirmed their sexual intercourse frequency decreased, and sexual satisfaction decreased in 76 (15.3%). A significant positive correlation was found in that those who disagreed with having a good sexual relationship tended to have severe stress (*p* < 0.001). Conclusion: There were increased levels of stress during the lockdown period, which impacted psychosexual health.

## 1. Introduction

In December 2019, unexpected cases of individuals with pneumonia caused by the novel coronavirus (COVID-19) were identified in Wuhan, China [1], and the virus’s propagation quickly escalated into a global health issue [2]. To prevent the virus from spreading uncontrollably, most countries have imposed limitations. Although social distance and other measures, such as the use of personal protective equipment (PPE), may help to restrict the spread of SARS-CoV-2 [3], they appear to have a negative impact on mental health [4]. Although COVID-19 is a new coronavirus strain, it has been linked to illnesses ranging from the common cold to more serious illnesses such as SARS and MERS [5]. Severe cases of the condition can result in cardiac and respiratory failure, as well as acute respiratory syndrome and death [6]. COVID-19, in addition to its physical consequences, has the potential to harm people’s mental health [7]. Indeed, the pandemic has resulted in a high incidence of mental health disorders in the general population, including acute stress, post-traumatic stress, anxiety, depression, irritability, insomnia, and decreased attention [4,8]. People are also more likely to fear becoming ill or dying, feeling helpless, and being stereotyped by others [9]. The epidemic has had a negative impact on public mental health, perhaps leading to psychiatric crises [10].

Married participants had a 40% lower risk of acquiring anxiety during COVID-19 lockdowns than unmarried participants, according to an Indian study [11]. However, the following instances illustrate that marriage/relationship quality appears to mitigate the association between marriage/relationship and mental health [12]. Relationship disharmony has been linked to an increased risk of mood and anxiety disorders, according to findings from a population-based study in the United States [13]. High marital quality has been linked to lower stress and depression, as well as lower blood pressure and more slow-wave sleep [14]. Stress can cause erectile dysfunction, which can make sexual activity difficult. For a long time, ED was thought to be primarily a psychological and distressing problem [15].

Because this topic is not widely discussed in the scientific literature, this research focuses on its possible hazards for male psychosexual health. A recent systematic review of 13 original studies on the impact of COVID-19 lockdown on psychological health and well-being was conducted, with some of the reviewed studies demonstrating an increased correlation between social media exposure and psychological issues such as sleep deprivation, as well as an increase in alcohol withdrawal and increased gaming behavior following the sudden lockdown. Only two studies have found evidence of poor psychosexual health [16].

So far, no study in Saudi Arabia has looked at the impact of COVID-19 lockdown stress on sexual activity, partner relationships, or psychosexual health in males who have been infected with the virus. As a result, the purpose of this observational study was to determine the intensity of stress during COVID-19 and the relationship between stress levels and partner relationships and sexual activity.

## 2. Methodology

A cross-sectional study was undertaken in Saudi Arabia for a period of two months between 1 December 2020 and 31 January 2021 via social media platforms. Using the Raosoft^®^ website, the minimal sample size was calculated to be 377, with a 5% margin of error and a confidence interval (percent) of 95 percent. A convenience sample strategy was used to obtain data from an online survey. A pilot study of 20 respondents was used to create, pretest, and validate the survey questionnaire. The questionnaire’s validity and intelligibility were evaluated by prominent medical and academic specialists. The survey was created using the Google Forms platform (Google LLC, Mountain View, CA, USA) and distributed over social media platforms such as Twitter and WhatsApp. The survey request came in the form of a tweet or a WhatsApp message. These communications explained the study’s objective and provided a link to it, as well as requested permission to participate. A cover page appeared once subjects clicked on the survey link, detailing the study’s title, purpose, and time required to complete the survey. We did not gather respondents’ contact information, such as email addresses, or demand registration for the purpose of secrecy. They were invited to click “start the survey” and begin answering the survey questions if they agreed to participate. It was entirely voluntary, and complete anonymity was guaranteed. The questionnaire was sent to a total of 871 people. The study included 497 males who had been infected with COVID-19 in the previous 6 months, were sexually active, and were above the age of 18. The demographic data, evaluation of the impact of COVID-19 infection on sexual activity and partner relationships, and evaluation of stress associated with COVID-19 during the pandemic were all included in the survey questionnaire. The questions used to assess the sexual relationship involved questions highlighting the relationship with partner, sexual desire, frequency of intercourse, frequency of masturbation, satisfaction level, use of pornography, use of condom, use of drugs (libido enhancing), and abnormal sexual behavior. A five-level scale of responses from strongly agree to strongly disagree was used.

To assess depression associated with COVID-19, we used the Depression Anxiety Stress Scale’s stress scale of 21 validated items (DASS-21) [17]. The DASS-21 is a collection of three self-report scales that are used to assess depression, anxiety, and stress. Permission to use the scale has been granted. The Arabic version of the DASS-21 was employed because our target sample included Arabic speakers, and it had previously been established as a trustworthy and accurate tool for measuring mental health status among Arabic speakers [18]. The DASS-21 stress scale is susceptible to persistent nonspecific arousal levels. It evaluates restlessness, anxious arousal, and being easily upset/agitated, irritable/over-reactive, and impatient. Because of the study’s goal, we simply used the DASS-21’s stress scale (7 items) and compared it to particular questions in our survey. The rating scale was as follows: 0—Did not apply to me at all; 1—Applied to me to some degree, or some of the time; 2—Applied to me to a considerable degree or a good part of time; and 3—Applied to me very much or most of the time. A set of three self-report scales known as the Depression, Anxiety, and Stress Scale—21 Items (DASS-21) is used to measure the emotional states of depression, anxiety, and stress. The three DASS-21 scales each have seven items that are broken down into subscales with related material. Dysphoria, hopelessness, devaluation of life, self-deprecation, lack of interest or involvement, anhedonia, and inertia are all evaluated by the depression scale. Autonomic arousal, skeletal muscle effects, situational anxiety, and subjective sensation of anxious affect are all measured by the anxiety scale. The persistent nonspecific arousal levels are sensitive to the stress scale. It evaluates issues with relaxation, nervousness, and a tendency to become easily disturbed or irritated, irritable or too sensitive, and impatient. The scores for the relevant questions are added up to determine the scores for depression, anxiety, and stress. The foundation of the DASS-21 is a dimensional rather than a categorical conception of psychological disorder. The assumption on which the DASS-21 development was based (and which was confirmed by the research data) is that the differences between the depression, anxiety, and stress experienced by normal subjects and clinical populations are essentially differences in degree. The DASS-21 therefore has no direct implications for the allocation of patients to discrete diagnostic categories postulated in classificatory systems such as the DSM and ICD. Recommended cut-off scores for conventional severity labels (normal, moderate, and severe) are as follows: NB Scores on the DASS-21 need to be multiplied by 2 to calculate the final score.

Statistical analysis of the survey’s quantitative data was performed using IBM SPSS version 20.0 statistical software (SPSS, Inc., Chicago, IL, USA). Frequencies and percentages were used to present the results. As needed, comparisons between survey variables were made using the Chi-squared test/Fisher’s exact test. The significance threshold was set at *p* ≤ 0.05. The Research Ethics Committee of Majmaah University in Saudi Arabia approved the study.

## 3. Results

The questionnaire was distributed to 871 subjects, of whom 497 (57%) subjects with a history of COVID-19 infection were recruited for the study, while the remaining subjects who reported no COVID-19 infection were excluded. Nearly 85% of the participants belonged to the age range of 18 to 39 years, with most (58%) belonging to the 18–29-year age group. More than half of the participants, 258 (51.9%), were married, and 254 (51.1%) had a bachelor’s degree. Of the respondents, 349 (70.2%) never smoked, while 144 (29%) were smokers; moreover, one drank alcohol (0.2%), while three (0.6%) used tobacco and alcohol (Table 1).

While assessing the rates of stress using the DASS-21 stress scale score among the participants with COVID-19 infection, it was found that 203 (40.8%) had severe stress scores (severe and extremely severe scores merged), while 131 (26.4%) had moderate stress scores (Figure 1).

Based on the level of agreement, the seven items of the DASS-21 stress scale showed that almost half, 236 (47.5%), agreed they found it difficult to wind down, while 144 (29%) tended to over-react to situations, 176 (35.4%) agreed that they were using a lot of nervous energy, about 167 (33.6%) found themselves becoming agitated, 160 (32.2%) found it difficult to relax, 138 (27.8%) were intolerant of anything that kept them from getting on with what were they doing, and, finally, 164 (33%) found themselves rather touchy (Table 2).

Table 3 depicts that during the COVID-19 pandemic lockdown, around 260 respondents (52.3%) reported that they or their partner met people with positive COVID-19 infection. Meanwhile, 268 (53.9%) reported that their partner lived with them in the same house during home isolation, and 193 (38.8%) reported that their partner was infected with COVID-19. In the six months prior to the study being conducted, 268 respondents (53.9%) did not have sexual relationships, while 229 (46.1%) did. Regarding fertility, 29 (5.8%) had been diagnosed with male infertility before the COVID-19 pandemic, while 468 (94.2%) did not report such a diagnosis. Furthermore, 226 (45.5%) had children, while 271 (54.5%) did not.

For extensive analysis, a comparison of the severity of the DASS-21 stress scale score results with respondents’ partner relationship and sexual activity status was assessed via Chi-squared test/Fisher’s exact test. For most infected respondents who answered “Yes”, the variables “met a person infected with COVID-19”, “partner lived in the same house during home isolation”, and “raising children” showed no significant association with the severity of stress levels, while the majority of infected respondents who answered “No” for variables “partner infected with COVID-19” and “no diagnosis of male infertility before the pandemic” showed a significant difference and were associated with low levels of stress (*p* = 0.004 and *p* < 0.001, respectively). Infected respondents who answered “No” for the variable “no sexual relationship in the last six months during the pandemic” also showed a significant difference and were associated with low levels of stress (*p* = 0.001) depicted in Table 4.

Table 5 depicts that during the COVID-19 pandemic lockdown, 84 (16.9%) agreed that their sexual desire decreased, 90 (18.1%) confirmed their sexual intercourse frequency decreased, sexual satisfaction decreased in 76 (15.3%), masturbation frequency increased in 89 (18%), pornography use increased in 59 (11.9%), condom use increased in 50 (10%), usage of oral sex-enhancing drugs increased in 35 (7%), and, finally, 37 (7.4%) reported an increase in the practice of abnormal sexual behaviors, such as multiple sexual partners.

An attempt was made to identify the association between DASS-21 stress scale scores and sexual activity during the pandemic lockdown period among the respondents with a history of COVID-19. A significant positive correlation was found in that those who disagreed tended to have severe stress (*p* < 0.001), as shown in Table 6.

## 4. Discussion

COVID-19’s rapid proliferation has caused anxiety in people around the world, leading to mental health problems in individuals [19,20]. As a result, in this tough, harmful, and unprecedented moment, it is critical to evaluate and acknowledge people’s mental states. Individuals may exhibit signs of psychosis, anxiety, trauma, suicidal thoughts, and panic attacks, according to evidence [21,22]. COVID-19 is a new and unknown virus, and its quick spread, high fatality rate, and uncertainty about the future can create anxiety and stress [23].

Gender was found to be a consistent predictor of psychological outcomes in recent Chinese studies [24,25,26] addressing the impact of COVID-19 on psychological health: females were more influenced than their male counterparts by psychological anguish. Higher levels of psychological distress were associated with female gender, negative affect, and detachment, according to data from Italy based on a self-report questionnaire utilizing the DASS-21. Furthermore, Mazza et al. found that having a COVID-19-infected friend increased depression and stress, whereas having a history of stressful situations and medical difficulties increased despair and anxiety [27].

Using the DASS-21 and Impact of Event Scale-Revised (IES-R), data from the early phases of the COVID-19 outbreak in Saudi Arabia indicated that one-fourth of respondents suffered moderate to severe psychological effects [28]: 23.6 percent experienced moderate to severe psychological effects of the outbreak, while 28.3 percent, 24 percent, and 22.3 percent reported moderate to severe depression, anxiety, and stress symptoms, respectively. There was no significant correlation with any of the DASS-21 subscales among the respondents who were quarantined and tested for COVID-19, which could be explained by the negative findings they obtained and the lack of symptoms during the quarantine period, leading to feelings of assurance [28]. Other recent prevalence data from Saudi Arabia used the Arabic version of the DASS-21 to assess psychological impacts among 1597 participants, with 12 percent reporting moderate to severe stress, and stress levels were significantly higher among females, younger respondents, and health care providers [29]. Furthermore, a recent study by Fang et al. found that a drop in sexuality among adult Chinese men was linked to despair and anxiety as a result of COVID-19, resulting in lower life satisfaction and quality [30]. Stress levels were higher among males infected with COVID-19 in our study, which used the DASS-21’s seven-item stress scale. In all, 67.2 percent of respondents said they were stressed, ranging from moderate (131; 26.4 percent) to severe (203; 40.8 percent).

Approximately 52.3 percent of responders said they or their partner met people with positive COVID-19 infection during the COVID-19 pandemic lockdown, and 53.9 percent of respondents with positive COVID-19 infection said their partner lived with them in the same residence during the home isolation period. Furthermore, 38.8% of respondents said they had been infected with COVID-19 through their spouse. A total of 53.9 percent of respondents stated they had no sexual relationships in the six months prior to the poll, while 46.1 percent claimed they had.

There was a statistically significant positive link between partner relationships and sexual relationships (p.001) in participants with positive COVID-19 in the previous six months, which continued if their spouse resided with them in the same house during home isolation (p.001). Furthermore, afflicted individuals who stated that their partners were not infected with COVID-19 had considerably lower stress levels (*p* = 0.004). Our findings differ from those of Fang et al. [30] and Mazza et al. [27]. Our respondents reported modest levels of stress related to sexuality and partner relationships, indicating that their sexual connections were not jeopardized during the lockdown. This could be owing to the Saudi government’s early implementation of specific preventative measures and heightened public knowledge about the pandemic, which appeared to have a protective effect on people’s overall health [31].

## 5. Conclusions

During the COVID-19 pandemic lockdown in Saudi Arabia, it was found that sexual activity along with satisfaction decreased, whereas there was an increase in masturbation and pornography use. It was also found that there was a rise in stress levels, which had an influence on psychosexual health. In the near future, more large-scale research could be expected.

## Figures and Tables

**Figure 1 healthcare-10-01663-f001:**
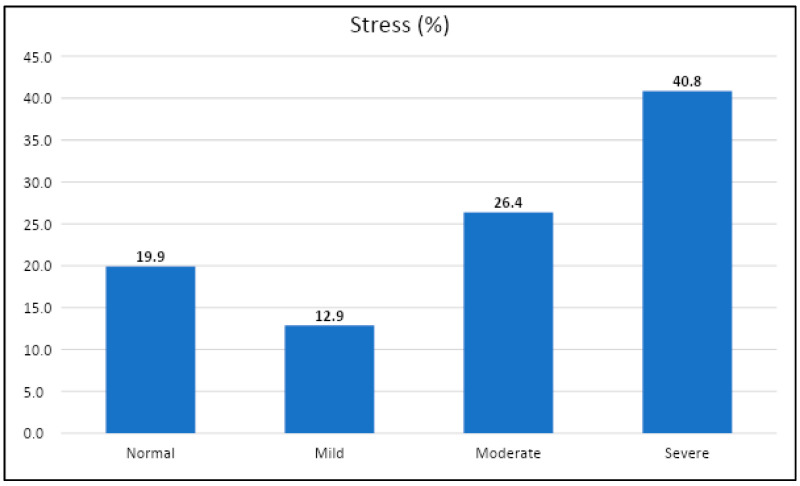
Level of stress using DASS-22 stress scale.

**Table 1 healthcare-10-01663-t001:** Participants’ demographic characteristics (*n* = 497).

Demographic Characteristics	*n* = 497	%
Age		
18–29 years	290	58.4
30–39 years	131	26.4
40–49 years	45	9.1
50–59 years	15	3.0
60–69 years	15	3.0
≥70 years	1	0.2
Current educational level		
High school degree and below	194	39.0
Bachelor’s degree	254	51.1
Masters and above	49	9.9
Current marital status		
Single	231	46.5
Married	258	51.9
Divorce	5	1.0
Widower	3	0.6
Addiction		
No smoking	349	70.2
Tobacco (hookahs/cigarettes/electric cigarettes)	144	29.0
Alcohol only	1	0.2
Both tobacco and alcohol	3	0.6

**Table 2 healthcare-10-01663-t002:** The seven items of the DASS-21 stress scale reported by responders with COVID-19 infection.

The Seven Items of the DASS-21 Stress Scale	*n* = 497	%
I found it hard to wind down.		
Agree	236	47.5
Neutral	75	15.1
Disagree	186	37.4
I tended to over-react to situations.		
Agree	144	29.0
Neutral	108	21.7
Disagree	245	49.3
I felt that I was using a lot of nervous energy.		
Agree	176	35.4
Neutral	84	16.9
Disagree	237	47.7
I found myself getting agitated.		
Agree	167	33.6
Neutral	88	17.7
Disagree	242	48.7
I found it difficult to relax.		
Agree	160	32.2
Neutral	89	17.9
Disagree	248	49.9
I was intolerant of anything that kept me from getting on with what I was doing.		
Agree	138	27.8
Neutral	102	20.5
Disagree	257	51.7
I felt that I was rather touchy.		
Agree	164	33.0
Neutral	78	15.7
Disagree	255	51.3

**Table 3 healthcare-10-01663-t003:** COVID-19 and partner relationship reported by responders during the pandemic lockdown. (*n* = 497).

Characteristics	Responses	Frequency	%Age
Contact with a COVID-19-infected person			
	Yes	260	52.3
	No	237	47.7
Residing with infected partner in the same house			
	Yes	268	53.9
	No	229	46.1
Partner infected with COVID-19			
	Yes	193	38.8
	No	304	61.2
History of sexual relationship in the last six months			
	Yes	229	46.1
	No	268	53.9
Any diagnosis of male infertility before the pandemic			
	Yes	29	5.8
	No	468	94.2
Has children			
	Yes	226	45.5
	No	271	54.5
Total		497	100.0

**Table 4 healthcare-10-01663-t004:** Correlation between the DASS-21 stress scale score with partner relationship of COVID-19 responders during pandemic lockdown.

DASS-21 Stress Scale Score										
	Normal		Mild		Moderate		Severe		Total	*p*-Value *
	*n*	%	*n*	%	*n*	%	*n*	%		
Have you or your partner met a person infected with the COVID-19?										
Yes	61	23.5%	37	14.2%	71	27.3%	91	35.0%	260	0.062
No	38	16.0%	27	11.4%	60	25.3%	112	47.2%	237	
After you get COVID-19 infection, during your home isolation, does your partner live with you in the same house?										
Yes	62	23.1%	33	12.3%	69	25.7%	104	38.8%	268	0.356
No	37	16.2%	31	13.5%	62	27.1%	99	43.2%	229	
Has your partner been infected with the COVID-19?										
Yes	41	21.2%	27	14.0%	65	33.7%	60	31.1%	193	0.004
No	58	19.1%	37	12.2%	66	21.7%	143	47.1	304	
In the last six months, have you had a sexual relationship?										
Yes	56	24.5%	35	15.3%	67	29.3%	71	31.1%	229	0.001
No	43	16.0%	29	10.8%	64	23.9%	132	49.1%	268	
Did diagnosed with male infertility before the pandemic?										
Yes	14	48.3%	5	17.2%	7	24.1%	3	10.3%	29	<0.001
No	85	18.2%	59	12.6%	124	26.5%	200	42.7%	468	
Currently, do you have children?										
Yes	43	19.0%	30	13.3%	67	29.6%	86	38.0%	226	0.382
No	56	20.7%	34	12.5%	64	23.6%	117	43.1%	271	

* Chi-square test.

**Table 5 healthcare-10-01663-t005:** Sexual behavior and partner sexual relationship during the COVID-19 pandemic lockdown period (*n* = 497).

	Strongly Agree		Agree		Neutral		Disagree		Strongly Disagree	
	*n*	%	*n*	%	*n*	%	*n*	%	*n*	%
My relationship with my partner is good	151	30.4	108	21.7	115	23.1	95	19.1	28	5.6
My sexual desire decreased	23	4.6	61	12.3	118	23.7	194	39.0	101	20.3
My sexual intercourse frequency decreased	31	6.2	59	11.9	136	27.4	184	37.0	87	17.5
My sexual satisfaction decreased	23	4.6	53	10.7	137	27.6	192	38.6	92	18.5
The masturbation frequency increased	32	6.4	57	11.5	94	18.9	200	40.2	114	22.9
The use of pornography increased	24	4.8	35	7.0	87	17.5	218	43.9	133	26.8
The use of condoms increased	18	3.6	32	6.4	82	16.5	207	41.6	158	31.8
Recreational use the oral sex enhancing drugs like PDE5-inhibitors increased	13	2.6	22	4.4	82	16.5	207	41.6	173	34.8
There are increase in my practice of abnormal sexual behaviors (example: multiple sexual partnerships)	17	3.4	20	4.0	73	14.7	201	40.4	186	37.4

**Table 6 healthcare-10-01663-t006:** Correlation between the DASS-21 stress scale score with sexual activity of COVID-19 responders during pandemic lockdown.

	The DASS-21 Stress Scale Score								Total	*p*-Value
	Normal		Mild		Moderate		Severe			
	*n*	%	*n*	%	*n*	%	*n*	%		
Sexual activity agreement level										
My relationship with my partner is good										
Strongly Agree	30	19.9%	16	10.6%	40	26.5%	65	43.0%	151	<0.001 *
Agree	24	22.2%	19	17.6%	39	36.1%	26	24.1%	108	
Neutral	28	24.3%	20	17.4%	36	31.3%	31	27.0%	115	
Disagree	12	12.6%	6	6.3%	11	11.6%	66	69.5%	95	
Strongly Disagree	5	17.9%	3	10.7%	5	17.9%	15	53.6%	28	
My sexual desire decreased										
Strongly Agree	10	43.5%	2	8.7%	3	13.0%	8	34.8%	23	<0.001 *
Agree	21	34.4%	16	26.2%	20	32.8%	4	6.6%	61	
Neutral	31	26.3%	18	15.3%	41	34.7%	28	23.7%	118	
Disagree	25	12.9%	22	11.3%	43	22.2%	104	53.6%	194	
Strongly Disagree	12	11.9%	6	5.9%	24	23.8%	59	58.4%	101	
My sexual intercourse frequency decreased										
Strongly Agree	11	35.5%	3	9.7%	8	25.8%	9	29.0%	31	<0.001 *
Agree	23	39.0%	15	25.4%	15	25.4%	6	10.2%	59	
Neutral	29	21.3%	26	19.1%	49	36.0%	32	23.5%	136	
Disagree	27	14.7%	15	8.2%	41	22.3%	101	54.9%	184	
Strongly Disagree	9	10.3%	5	5.7%	18	20.7%	55	63.2%	87	
My sexual satisfaction decreased										
Strongly Agree	8	34.8%	4	17.4%	5	21.7%	6	26.1%	23	<0.001 *
Agree	25	47.2%	12	22.6%	12	22.6%	4	7.5%	53	
Neutral	28	20.4%	26	19.0%	47	34.3%	36	26.3%	137	
Disagree	30	15.6%	17	8.9%	48	25.0%	97	50.5%	192	
Strongly Disagree	8	8.7%	5	5.4%	19	20.7%	60	65.2%	92	
The masturbation frequency increased										
Strongly Agree	15	46.9%	3	9.4%	4	12.5%	10	31.3%	32	<0.001 *
Agree	25	43.9%	13	22.8%	11	19.3%	8	14.0%	57	
Neutral	18	19.1%	15	16.0%	36	38.3%	25	26.6%	94	
Disagree	25	12.5%	24	12.0%	51	25.5%	100	50.0%	200	
Strongly Disagree	16	14.0%	9	7.9%	29	25.4%	60	52.6%	114	
The use of pornography increased										
Strongly Agree	12	50.0%	4	16.7%	2	8.3%	6	25.0%	24	<0.001 *
Agree	17	48.6%	6	17.1%	8	22.9%	4	11.4%	35	
Neutral	18	20.7%	19	21.8%	32	36.8%	18	20.7%	87	
Disagree	33	15.1%	23	10.6%	55	25.2%	107	49.1%	218	
Strongly Disagree	19	14.3%	12	9.0%	34	25.6%	68	51.1%	133	
The use of condoms increased										
Strongly Agree	9	50.0%	2	11.1%	4	22.2%	3	16.7%	18	<0.001 *
Agree	12	37.5%	8	25.0%	8	25.0%	4	12.5%	32	
Neutral	13	15.9%	13	15.9%	30	36.6%	26	31.7%	82	
Disagree	38	18.4%	26	12.6%	46	22.2%	97	46.9%	207	
Strongly Disagree	27	17.1%	15	9.5%	43	27.2%	73	46.2%	158	
Recreational use the oral sex enhancing drugs like PDE5-inhibitors increased										
Strongly Agree	6	46.2%	3	23.1%	2	15.4%	2	15.4%	13	<0.001 **
Agree	12	54.5%	4	18.2%	3	13.6%	3	13.6%	22	
Neutral	12	14.6%	14	17.1%	34	41.5%	22	26.8%	82	
Disagree	36	17.4%	27	13.0%	49	23.7%	95	45.9%	207	
Strongly Disagree	33	19.1%	16	9.2%	43	24.9%	81	46.8%	173	
There are increase in my practice of abnormal sexual behaviors (example: multiple sexual partnerships)										
Strongly Agree	9	52.9%	2	11.8%	3	17.6%	3	17.6%	17	<0.001 **
Agree	9	45.0%	7	35.0%	4	20.0%	0	0.0%	20	
Neutral	15	20.5%	8	11.0%	29	39.7%	21	28.8%	73	
Disagree	34	16.9%	24	11.9%	50	24.9%	93	46.3%	201	
Strongly Disagree	32	17.2%	23	12.4%	45	24.2%	86	46.2%	186	
Total	99	19.9%	64	12.9%	131	26.4%	203	40.8%	497	

* Chi-square test. ** Fisher exact test.

## Data Availability

Data presented in this study are available in this article. Further data sharing is not available.

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
