# Peer review of "Effect of COVID-19-Induced Stress among Males on the Partner Relationship and Sexual Activity during COVID-19 Pandemic: A Cross-Sectional Study"

_healthcare, 2022, doi:10.3390/healthcare10091663_

Round 1
Reviewer 1 Report
The work is very interesting and systematically written. The paper is reviewable and easy to read. I suggest that it be accepted with minimal corrections.
1. The numbers in parentheses come before the period
2. In the methodology, describe in more detail the questions and the method of collecting data on sexual behavior
3. In the results, it is not necessary to comment on each question from the DASS scale individually. Throw out those tables.
I congratulate the authors on their efforts. It was a pleasure to be a reviewer of this paper
Author Response
Dear sir,
Thanks a lot for your valuable comments. We have tried addressing and hence changing the required fields as per your suggestions.
Regards

Reviewer 2 Report
The article addresses the important issue of the importance of the COVID-19 pandemic for the quality of partner relationships and sexual activity. While the issue of the impact of the pandemic on relationships in close relationships has already been addressed in various studies, the issue of sexual activity has been addressed much less frequently. Below are comments that I hope will allow the Authors to refine the manuscript.
I would have expected a slightly more extensive introduction, which would include a stronger justification for the research problem. It would be good to clearly formulate the problem and questions and/or research hypotheses.
The tools used should be described in more detail (give examples of items, describe the scale on which the respondents answered the questions, explain in what form the final result was obtained). Psychometric characteristics of the tools should also be given. It would also be good to justify the reason for choosing only one of the subscales. It would be appropriate to briefly justify the questions posed, especially the question: Did diagnosed with male infertility before the pandemic? It would also be good to justify the selection of people for the study group and the fact that only men participated in the study.
It is not clear why the study, which aimed to assess the quality of relationships, involved people referred to as singles. Did the people referred to as singles have a partner, were they in a relationship? If not, then all questions about the partner seem unfounded, as well as the assessment of the quality of the relationship. However, if they had a partner, can they be called singles? The text does not indicate whether only people with a partner (including those who are not yet married) participated in the research. Is it known whether the subjects undertook sexual activity at all/underwent sexual initiation? How long did it take from the time the subjects were infected with COVID-19 to the time they participated in the study? It would be good to discuss the results of the research in more detail.
The discussion lacked an attempt at interpretation and reflection on the significance of the obtained results. The grounds for the proposal are not clear: There was a statistically significant positive link between partner relationships and sexual relationships (p.001) in the participants with positive COVID-19 in the previous six months, which continued if their spouse resided with them in the same house during home isolation (p.001). What is the measure of partner and sexual relationships, and does this conclusion apply to people who have a partner and did not live with them while they were infected, or also to people who are single? The authors state: Stress levels were higher among males infected with COVID-19 in our study, which used the DASS-21's seven-item stress scale. In all, 67.2 percent of respondents said they were stressed, ranging from moderate (131; 26.4 percent) to severe (203; 40.8 percent). It is unclear whether the Authors assume that stress levels are higher in those infected than in people who were not infected (such people have not been studied), or higher than in other studies cited, or simply state that most of the subjects declare elevated stress levels.
It would be good to point out the limitations of own research and the directions of further research, as well as clearly formulate conclusions from own research.
Author Response
Dear sir,
Thanks a lot for your valuable comments. We have tried addressing and changing the required fields as per your suggestions.
Regards

Reviewer 3 Report
The abstract is well written. The introduction has almost a good description. References are linked to the references in the introduction. The research method lacks an instrument for assessing the quality of sexual life. It is not enough just to have tools that affect the mental state. It was more advisable to have instruments for the evaluation of the quality of sexual life. The results are very good. The discussion section has data relevant to arguments. Congratulations to the authors.
Author Response

(The authors gave the same response as above.)
